

# A national-scale seasonal hydrological forecast system: development and evaluation over Britain

Victoria A. Bell[1], Helen N. Davies[1], Alison L. Kay[1], Anca Brookshaw[2,3] and Adam A. Scaife[3,4]

[1] Centre for Ecology & Hydrology, Wallingford, Oxfordshire, OX10 8BB, UK
[2] ECMWF, Shinfield Park, Reading, RG2 9AX, UK
[3] Met Office Hadley Centre, FitzRoy Road, Exeter, Devon, EX1 3PB, UK
[4] College of Engineering, Mathematics and Physical Sciences, University of Exeter, Exeter, United Kingdom, EX4 4QF

*Correspondence to*: V.A.Bell (vib@ceh.ac.uk)

**Abstract.** Skilful winter seasonal predictions for the North Atlantic circulation and Northern Europe have now been
demonstrated and the potential for seasonal hydrological forecasting in the UK is now being explored. One of the techniques
being used combines seasonal rainfall forecasts provided by operational weather forecast systems with hydrological
modelling tools to provide estimates of river flows up to a few months ahead.

The work presented here shows how spatial information contained in a distributed hydrological model typically requiring
high resolution (daily or better) rainfall data can be used to provide an initial condition for a much simpler forecast model
tailored to use low-resolution monthly rainfall forecasts. Rainfall forecasts ("hindcasts") from the GloSea5 model (1996 to
2009) are used to provide the first assessment of skill in these national-scale flow forecasts. The skill in the combined
modelling system is assessed for different seasons and regions of Britain, and compared to what might be achieved using
other approaches such as use of an ensemble of historical rainfall in a hydrological model, or a simple flow persistence
forecast. The analysis indicates that only limited forecast skill is achievable for Spring/Summer seasonal hydrological
forecasts, however, Autumn/Winter flows can be reasonably well forecast using (ensemble mean) rainfall forecasts based on
either GloSea5 forecasts or historical rainfall (the preferred type of forecast depends on the region). Flow forecasts using
ensemble mean GloSea5 rainfall perform the most consistently well across Britain, and provide the most skilful forecasts
overall at the 3-month lead time. Much of the skill (64%) in the 1-month ahead seasonal flow forecasts can be attributed to
the hydrological initial condition (particularly in regions with a significant groundwater contribution to flows), whereas for
the 3-month ahead lead time, GloSea5 forecasts account for ~70% of the forecast skill (mostly in areas of high rainfall to the
North and West) and only 30% of the skill arises from hydrological memory (typically groundwater-dominated areas).

Given the high spatial heterogeneity in typical patterns of UK rainfall and evaporation, future development of skilful
*spatially distributed* seasonal forecasts could lead to substantial improvements in seasonal flow forecast capability,
benefitting practitioners interested in predicting hydrological extremes, not only in the UK, but potentially across Europe.




## 1 Introduction

A series of low-pressure systems crossing Britain in Winter 2015/16 resulted in some of the most widespread and severe flooding witnessed in the UK, with several rivers in the north of Britain recording their highest ever flows and thousands of properties flooded (Centre for Ecology & Hydrology 2016). Repairs to damaged homes, businesses and flood defences were

required, and procedures for forecasting and mitigating the floods are understandably being examined. In Britain, a perceived lack of skill in seasonal weather forecasts in extratropical regions beyond a lead time of 1 month (Lavers et al. 2009, *Arribas et al.* 2011) has until recently discouraged the development of routine seasonal hydrological forecasts using climate model output. However, the potential for seasonal hydrological forecasting in the UK is now being explored. Various seasonal forecast systems now provide skilful forecasts out to a few months ahead (e.g. MacLachlan et al. 2015,

Athanasiadis et al. 2014), allowing for some form of skilful dynamical hydrological forecast. As well as using climate model output, others are investigating statistical relationships between large-scale North Atlantic climate indices (such as the North Atlantic Oscillation) and seasonal rainfall or river flow anomalies (Lavers et al. 2010a, 2010b; Macgregor and Phillips 2004; Svensson and Prudhomme 2005; Wedgbrow et al. 2002; Wilby 2001, Svensson et al. 2015), and these can provide increased skill when large scale patterns dominate regional rainfall (Scaife et al. 2014).

A recent review of seasonal hydrological forecasting methods using climate model output by Yuan et al. (2015) highlighted the dependence of predictive skill on both the large scale climate drivers and the local hydrological initial condition (HIC), which for some regions can persist for several months. The relative importance of initial conditions and boundary forcing (the meteorological forecast) on the skill of seasonal hydrological prediction has been examined by a number of authors, for

example a study of skill in forecasting mean seasonal river flows across Europe concluded that much of the skill could be attributed to correct hydrological initial conditions, rather than the weather forecast (Bierkens and van Beek, 2009). In a UK-based study, also using seasonal forecasts, Svensson et al. (2015) identified a geographical complementarity in regional seasonal hydrological predictability, noting that predictability in river flows in southern and eastern Britain derived primarily from hydrological memory of antecedent conditions, and from meteorological predictability (predictions of the atmospheric

circulation over the North Atlantic at the seasonal timescale) in northern and western areas. They were able to generate skilful hydrological forecasts for river flows using the large scale atmospheric circulation which governs much of UK winter (December to February) rainfall, and November initial conditions.

Advances in the performance of operational seasonal forecast systems such as the Met Office GloSea5 system (MacLachlan

et al. 2015) are now encouraging the development of hydrological forecasting systems that can make best use of these more skilful seasonal forecasts. In the UK, the recently developed Hydrological Outlook UK (HOUK) provides an insight into future hydrological conditions nationwide. It describes likely trajectories for river flows and groundwater levels on a monthly basis, with particular focus on the next one and three months. A number of techniques are used to project forwards

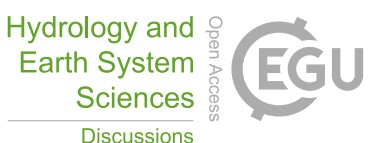

from the current state, and results from these are used to produce a summary including a highlights map. Prudhomme et al (under review) summarises the range of techniques used in the production of the HOUK, which encompass schemes using historical river flow analogues, ensembles of historical sequences of observed climate, and ensembles of seasonal rainfall forecasts.

For the latter approach, which is explored in more detail here, seasonal rainfall forecasts provided by the Met Office model GloSea5 are combined with hydrological modelling tools to provide estimates of hydrological conditions up to a few months ahead. The hydrological modelling follows on from an approach to seasonal forecasting developed by Bell et al. (2013), which used a distributed hydrological model driven by observations to provide the hydrological initial condition, and a

monthly 'water-balance anomaly' model to estimate sub-surface water storage over the next 1 to 3 months as perturbations from the initial state, driven by Met Office seasonal rainfall forecasts. Forecasting UK-wide monthly-mean river flow is less straightforward than forecasting subsurface water storage, as river flow is a spatial and temporal integrator of local-scale runoff production, arising from a combination of antecedent storage and the partitioning of effective rainfall between surface /subsurface runoff and storage. This task can arguably be undertaken by a fully configured grid-based hydrological model,

maintaining a continuous local water balance and using daily or sub-daily spatial rainfall estimates as input. However, seasonal rainfall forecasts do not provide detailed weather information at this resolution and would typically require spatio-temporal downscaling to achieve realistic national estimates of river flow. While rainfall downscaling is relatively straightforward for a particular location or catchment, turning national-scale monthly rainfall forecasts into pixel-scale daily rainfall would require an ensemble approach based on either a weather generator or historical analogues, thus generating

ever-greater numbers of ensemble flow forecasts.

The alternative approach explored here is to place the greatest emphasis on the hydrological initial condition provided by an up-to-date model, while simplifying the generation of hydrological forecasts through use of a temporally-coarse water balance model with less dependence on high resolution weather information. A monthly-resolution forecast model provides

additional benefits by reducing the computational overhead of the use of a rainfall forecast ensemble. This scheme is used to provide regional-scale estimates of the river flows over the coming months, and work presented here examines the skill of these forecasts for Britain, for geographical regions, for particular seasons, and at 1- and 3-month lead times.

## 2 Models and Methods

### 2.1 Hydrological initial condition

Grid-to-Grid (G2G) is a spatially-distributed hydrological model, which is generally configured to a 1km$^2$ grid across Britain, with a 15-minute time-step, and underpinned by digital spatial datasets of topography, soil/geology and land cover.



A detailed description of G2G is presented in Bell *et al.* (2009), with a brief overview of the model's subsurface (soil and groundwater) storage formulation provided in Bell et al. (2013). Input to the model consists of gridded time-series of precipitation and potential evaporation (PE) derived from observations, numerical weather prediction or regional climate models. Model output can be in the form of area-wide, gridded time-series of river flows, runoff and soil-moisture, or time-
series of river flows at gauged or ungauged locations. Applications of the model include both continuous simulation of river flows in a changing climate (Bell *et al.*, 2009, 2016) and real-time flood forecasting (Moore *et al.*, 2006; Cole and Moore, 2009).

For production of the HOUK, G2G is run continuously over several years to produce an estimate of the most recent
hydrological condition across Britain, from which an estimate is made of the current depth of subsurface water storage. The G2G requires gridded time-series of rainfall and PE. Daily precipitation data on a 5km grid, provided by the Met Office for 1958–present (Perry et al. 2009), were used at the 15-minute G2G time-step by equally spreading them throughout the day, and downscaled to 1km using a spatial weighting based on 1km Standard Average Annual Rainfall data for 1961–1990 (Bell et al. 2007). Monthly PE data on a 40km grid from MORECS (Hough and Jones 1997) were spread equally through the
month and applied equally to each 1km box within each 40km square. Here, the depth of sub-surface water storage, $S$, consists of the sum of the unsaturated soil, $V$, and the groundwater, $V_g$, storages. The depth of water in groundwater storage arises from the balance between recharge and groundwater outflow over long periods, and while it is unlikely to correspond directly to a groundwater level observation, it can provide an indication of whether storage in the saturated zone is greater or less than the long-term average.

**2.2 Water-balance model for flows**

Following Bell et al. (2013) the continuity equation can be used to express change in total subsurface water storage, $S$, as a balance between input precipitation $P$ and outputs through actual evaporation $E$ and net outflow (surface and sub-surface runoff) per unit area $Q$, so $\mathrm{d}S/\mathrm{d}t = P - E - Q$, where all quantities are expressed in water depth (mm) over a model grid-cell. In terms of discrete months, if $S_m$ and $S_{m+1}$ represent the storage at the end of months $m$ and $m+1$, and if
$P_{m+1}, E_{m+1}$ and $Q_{m+1}$ denote mean rainfall, evaporation and runoff over the month $m+1$,

$$S_{m+1} \cong S_m + P_{m+1} - E_{m+1} - Q_{m+1} \qquad (1)$$

At a monthly time-step it is assumed here that daily/sub-daily changes in rainfall, storage, and river flows can be neglected and gross simplifying assumptions can be made as to whether excess effective rainfall is stored in the subsurface or released via runoff from saturated pixels. Storage of water in each pixel is assumed to vary between $S_{min}$ and $S_{max}$, the historical
minimum and maximum G2G-simulated storage of each 1km pixel's sub-surface water store respectively. By combining the




current storage $S_m$ as estimated by the G2G at the forecast time origin with monthly seasonal forecasts of $P_{m+1}$ and $E_{m+1}$, corresponding forecasts of storage, runoff and flow can be produced as follows:

1. For a discrete month, *m*, an initial estimator of the storage in each pixel at the end of the following month *(m+1)* can be given by

$$\hat{S}^*_{m+1} \approx S_m + P_f - E_f \qquad (2)$$

where $P_f$ and $E_f$ are seasonal forecasts of mean monthly rainfall and actual evaporation, and the * indicates an initial (as opposed to final) estimator.

2. The initial estimator for forecast storage $\hat{S}^*_{m+1}$ (Eq. 2) neglects forecast runoff $Q_{m+1}$ which can be significant, but is less easy to forecast directly than storage as its magnitude will depend on a number of factors including soil properties, storage, effective rainfall and topography. Typically in hydrological models, runoff is estimated through a relationship between incoming effective rainfall and antecedent soil-moisture and sub-surface water storage, and for the seasonal forecasting application considered here, runoff is also estimated through a relationship of the form $Q \simeq f(S)$. For the national-scale application required for the HOUK encompassing a wide range of soils, geology and catchment characteristics, a very simple empirical relationship is assumed:

$$\hat{Q}_{m+1} \simeq \hat{S}^*_{m+1} \frac{\bar{Q}_{m+1}}{\bar{S}_{m+1}} . \qquad (3)$$

Here, mean monthly runoff in month *m+1* is estimated in terms of the forecast storage $\hat{S}^*_{m+1}$ scaled by the ratio between G2G model-derived estimates of long-term mean runoff $\bar{Q}_{m+1}$ and storage $\bar{S}_{m+1}$ for the month (1962 to 2010).

3. Replacing the unknown $Q_{m+1}$ in Eq. (1) with its estimator $\hat{Q}_{m+1}$ yields an improved estimate of $S_{m+1}$:

$$\hat{S}_{m+1} \approx (S_m + P_f - E_f)\left(1 - \frac{\bar{Q}_{m+1}}{\bar{S}_{m+1}}\right) \approx \begin{cases} \hat{S}^*_{m+1}\left(1 - \frac{\bar{Q}_{m+1}}{\bar{S}_{m+1}}\right), & \text{for } \frac{\bar{Q}_{m+1}}{\bar{S}_{m+1}} < 1 \\ 0, & \text{else.} \end{cases} \qquad (4)$$

Eqs (1), (3) and (4) form the basis of the water balance model (WBM), which considers two situations according to whether the forecast sub-surface storage in the pixel is saturated:

- For saturated pixels, defined as $\hat{S}_{m+1} \geq S_{max}$, further excess rainfall cannot be accommodated as subsurface storage and is instead assumed to contribute directly to surface runoff and river flows. Then storage at the end of the next month $S_{m+1} = S_{max}$ and, re-arranging Eq. (1), $Q_{m+1} \approx S_m + P_f - E_f - S_{max}$



- For unsaturated pixels, defined as $\hat{S}_{m+1} < S_{max}$, excess rainfall is assumed to contribute to both sub-surface storage and runoff, and forecasts of these variables are estimated from Eqs (3) and (4) respectively. For locations where $\frac{\bar{Q}_{m+1}}{\bar{S}_{m+1}} \geq 1$, to maintain continuity, $S_{m+1} = 0$, and $Q_{m+1} = S_m + P_f - E_f$.

Runoff totals (mm) for each grid-square in each region are converted to monthly mean river flows (m³s⁻¹) through lateral transfer of upstream runoff from each catchment to the catchment outlet for every river grid-cell, using the 1km flow directions identified for the kinematic wave routing implemented in the G2G Model (Davies and Bell 2008, Bell et al. 2009). WBM flows for every 1km river location are scaled with respect to historical mean WBM flow (1962 -2010) and these standardised flows are averaged to provide a mean value for each of 17 geographic regions (Figure 1b). The coarse spatial
resolution of the input rainfall forecasts has discouraged the development of river flow forecasts at a 1km resolution, and production of regional scale forecasts (in preference to national scale) is viewed as a pragmatic compromise.

**2.3 Seasonal rainfall forecasts**

The long range meteorological forecasts used here were produced using the Met Office Global Seasonal forecast System (MacLachlan *et al.* 2015) and consist of a multi-member ensemble of UK-average (i.e. spatially uniform) monthly total
rainfall forecast for the next month, available at the start of each month. The climate model at the core of this forecast system has atmospheric resolution of 0.83 degrees longitude by 0.55 degrees latitude, 85 quasi-horizontal atmospheric levels and an upper boundary at 85km near the mesopause to represent stratospheric processes which are important for winter forecasts (Scaife et al. 2016). The ocean resolution is 0.25° globally in both latitude and longitude with 75 quasi-horizontal levels. This ocean resolution is necessary to reduce key biases in the ocean and atmosphere and give a realistic winter atmospheric
blocking climatology in the model (Scaife et al. 2011). A multi-member ensemble of 1- and 3-month rainfall forecasts (mm/day) was run for each season in the period 1996 to 2009 with lagged start dates centred on 1ˢᵗ February, 1ˢᵗ May, 1ˢᵗ August and 1ˢᵗ November. 12 ensemble members were available for forecasts starting in August and February and 24 for those starting in May and November. Thus ensembles of 1-month ahead forecasts are provided for December, March, June and September, and 3-month ahead forecasts for Winter (DJF), Spring (MAM), Summer (JJA) and Autumn (SON).
Members from the same start date differ only by stochastic physics. Initial atmospheric and land surface data were taken from ERA interim observational reanalyses and initial conditions for the global ocean and sea ice concentration were from the FOAM data assimilation system (Blockley *et al.* 2014).

The spatially-uniform rainfall forecasts present a dilemma for hydrological modellers who typically require high spatial and
temporal resolution weather information to estimate a water balance and represent the highly spatially and temporally variable nature of streamflow. An ensemble of mean UK rainfall forecasts provides no information on whether the rainfall is more likely to occur in the North or South, however, it does provide some indication of whether the rainfall totals will be





higher or lower than the climatological (long-term) mean. Such spatially-uniform forecasts will be unable to provide the spatial heterogeneity observed in UK rainfall and would under/overestimate rainfall in Northern/Southern regions if used directly. Instead, the rainfall forecasts $P_f$ are converted to spatially uniform rainfall anomalies, $a = P_f - \bar{P}_f$ (mm) relative to the GloSea5 estimate of climatological mean rainfall ($\bar{P}_f$). A spatially distributed UK monthly rainfall amount, $P^*$, is then

calculated as $P^* = \frac{P_{ij}}{\bar{P}}(\bar{P} + a)$, where $\bar{P}$ and $P_{ij}$ are the UK-mean and the local (1km pixel) monthly mean rainfall (1971-2000) respectively.

To produce the 3-month ahead flow forecasts using the GloSea5 hindcast dataset, either sequential monthly rainfall forecasts or a 3-month mean rainfall forecast were available for use as input, thus for Winter (DJF), forecasts were available for D, J

and F separately and the 3-month (DJF) mean. In the analysis that follows forecast skill has been assessed using both temporal resolutions of rainfall forecast, but as the results are very similar, only results for the mean 3-month ahead forecast are presented. This is consistent with the monthly Hydrological Outlook for which forecasts at lead times of 1- and 3-months only are available. Disaggregation of the 3-month ahead forecast into monthly rainfall amounts is achieved through distributing the 3-month rainfall forecast anomaly (relative to the rainfall forecast climatological mean) between the 3

individual months according to the ratios of their UK mean SAAR (1962 to 2010).

### 2.4 Seasonal flow forecasts

To produce seasonal flow forecasts (hindcasts), the water balance model for flows (Section 2.2) is initialised with the most recent G2G estimate of sub-surface water storage (Section 2.1). GloSea5 seasonal rainfall forecasts (1- and 3-month ahead, Section 2.3) are applied alongside climatological monthly mean actual evaporation (AE) estimated from a long Grid-to-Grid

model run (1962 to 2010).

While the skill of a single set of forecasts can be compared to observations using measures such as the Pearson correlation coefficient, the performance of an ensemble of seasonal flow forecasts can more easily be assessed using the relative operating characteristic (ROC) skill score (Kharin and Zwiers 2003), used widely for probabilistic weather forecast

verification. For ensembles, the ROC is a curve that indicates the relationship between hit rate and false alarm rate as different sorted ensemble members are used as decision thresholds. The ROC is commonly summarized through the integrated area under the curve, $AUC$, using $S_{ROC}=2*AUC-1$: A perfect forecast has $S_{ROC}=1$ ($AUC=1.0$), while forecasts with no skill have $S_{ROC}<=0$ ($AUC<=0.5$). The scores are calculated separately for each of three severity bands (below normal, $0 - 28\%$; normal, 28-72%; above normal, 72-100%), by ranking standardised river flow forecasts for the 17 geographical

regions of Britain in relation to simplified percentile ranges of historical flow estimates for each month based on 49 years of WBM simulated flows (1962–2010). The relatively wide bands were selected to agree with that used by both the rest of the



HOUK methods, and by the Hydrological Summary produced by the National Hydrological Monitoring Programme (Dixon et al. 2013), and serve to highlight when flows are unusually high or low.

To assess the importance of various factors involved in the seasonal flow forecasts, the performance of four alternatives is compared:

(a)  WBM with GloSea5 rainfall forecasts and the most recent G2G HIC ("GloSea5+HIC");

(b)  WBM with GloSea5 rainfall forecasts and a G2G historical mean HIC (1962-2010) ("GloSea5+avHIC");

(c)  WBM with an historical observed rainfall ensemble (49 members, 1962-2010) and the G2G HIC ("Clim+HIC"); and

(d)  Flow persistence with the G2G HIC ("Pers").

Comparing (b) to (a) gives an idea of relative contribution of the HIC to forecast skill, while comparing (c) to (a) gives an idea of the relative contribution of GloSea5 rainfall to forecast skill. Flow persistence (d) (carrying the most recent flow anomaly forward to the next 1- and 3-months) provides a much simpler form of forecast, for overall comparison. The skill of WBM with the G2G historical mean HIC and the ensemble of historical observed rainfall was also assessed but, as would be expected, the ensemble of forecasts had zero skill ($S_{ROC}=0$) and for brevity have been excluded from the analysis. Performance results for the remaining four alternatives are presented in Section 0.

## 3 Results

### 3.1 Assessment of water balance model for flows

The performance of the WBM to produce flow forecasts was assessed at a regional scale for the period January 1962 to December 2010 using observed gridded rainfall inputs (i.e. assuming a perfect rainfall forecast) and monthly mean AE (1962-2010) from the G2G, and initialising the WBM each month with the most recent G2G HIC. The resulting output, consisting of a temporal sequence of fixed lead-time, 1- and 3-month ahead regional flow forecasts, was compared to G2G regional mean monthly flows derived from a continuous simulation from 1962-2010 driven by observed (spatially distributed) daily 5km gridded rainfall observations and MORECS PE as input (described in Section 2.1). A comparison with measured river flows at individual sites across Britain has not been undertaken because the WBM has been developed to provide regional monthly flows, and observed mean flows are not available at a regional scale.





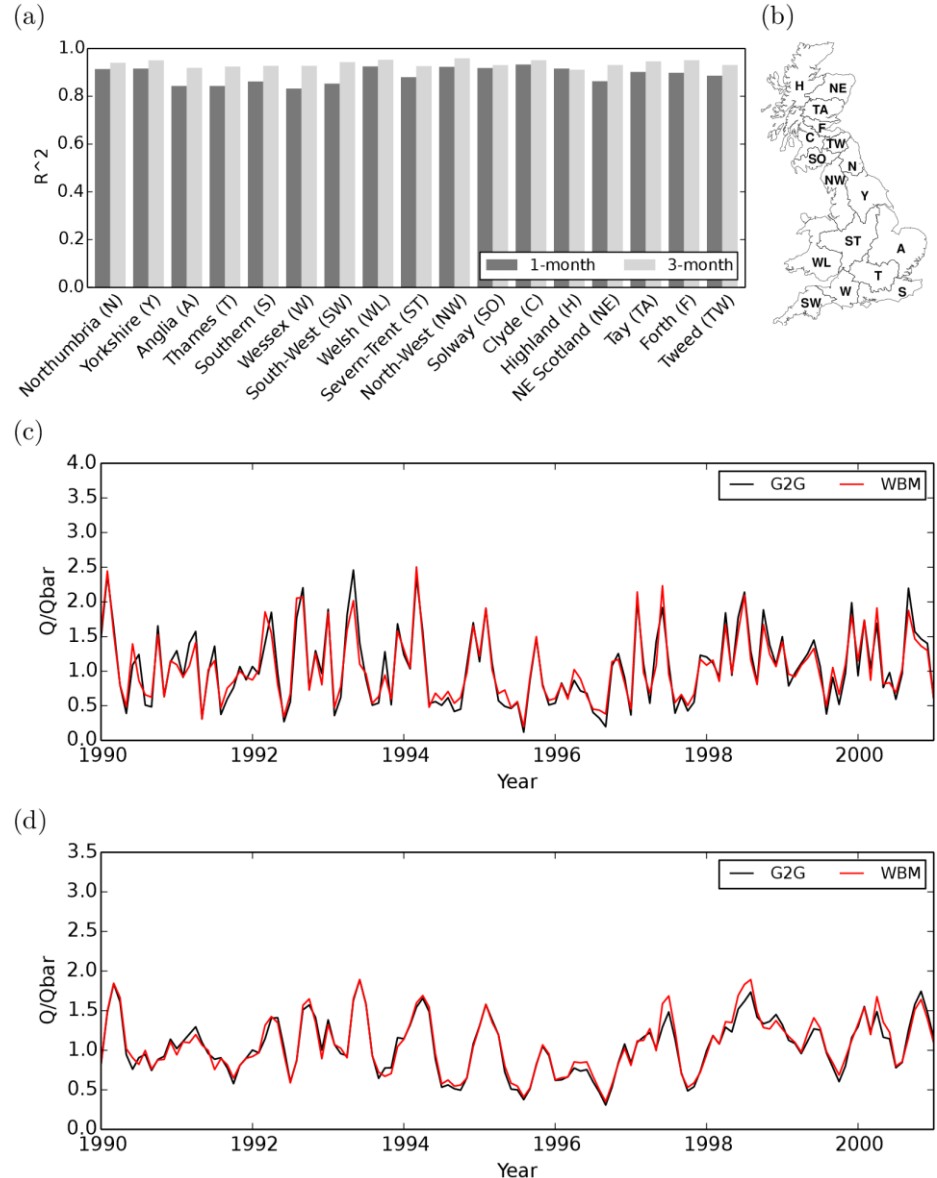

**Figure 1 (a) Regional WBM performance (1-month and 3-months ahead) compared to G2G output for 17 regions, in terms of the (Pearson) $r^2$. The regions are shown in (b). Monthly time-series of (c) 1-month and (d) 3-month ahead forecasts for the period January 1990 to December 2000: mean G2G (black) and WBM (red) simulated flows for the Forth region.**

Figure 1a provides a summary of WBM forecast performance in terms of the (Pearson) $r^2$ at the regional scale when compared to G2G output. For all 17 regions, the 1-month ahead WBM simulates more than 80% of the variability in G2G flows, and for 8 regions (typically upland regions highly responsive to rainfall), it explains more than 90%. The 3-month ahead forecasts all explain more than 90% of the variability. By way of example, Figure 1c and d show modelled regional





mean river flows for the Forth region (which has the median model performance), illustrating how closely the 1- and 3-month ahead WBM forecasts match continuous simulation G2G regional mean flows. Regional flows are estimated as the regional mean of $Q/\overline{Q}$ at every location (1km pixel) for which $\overline{Q} > 0.05 \, m^3 s^{-1}$. The division by $\overline{Q}$ enables equal weighting for upstream and downstream river locations.

## 3.2 Assessment of seasonal flow forecasts

An assessment of model skill using the $S_{ROC}$ skill score has been undertaken for Britain as a whole, for 17 regions, two lead-times, four forecast starting points (seasons). A skill assessment should ideally take into account all these factors, and although the average performance measure over all areas shown in Figure 2a disguises the complexity in regional response and forecast model performance at different times of year, it does immediately highlight the utility of using the HIC with rainfall ensembles (GloSea5 or historical) over use of an average HIC or flow persistence. It is important to note that although skill scores improve with the number of ensemble members (Scaife et al., 2014), for the skill assessments here, the ensemble size varies: the historical rainfall ensemble has 49 members, while the rainfall forecast ensemble has 12 or 24 members for autumn/summer and summer/winter respectively, Thus the forecast rainfall skill scores may be an underestimate of the real-time skill (operational GloSea5 forecasts have 42 members). At the 1-month lead time the WBM with G2G HIC driven by an historical rainfall (climatology) ensemble performs best, while for the longer 3-month lead time, the WBM with G2G HIC driven by either historical or GloSea5 rainfall perform similarly. Persistence forecasts (Pers) or use of an average HIC are not recommended.

When the overall performance scores shown in Figure 2a are split between seasons, utility of GloSea5 forecasts in September/Autumn and December/Winter becomes apparent (Figure 2b,c: GloSea5+HIC). For forecasts that use the HIC, use of an ensemble of historical rainfall provides some skill (S_ROC > 0) across all seasons, particularly at the 1-month lead time, but use of GloSea5 rainfall forecasts is more skilful in autumn (yellow bars), and also in winter (blue bars) at the 3-month lead time. There is little skill in summer flow forecasts (red bars) whatever type of rainfall forecasts is used, with the best 1-month ahead forecast performance achieved using historical rainfall, and best 3-month ahead forecasts from persistence of current flow conditions or historical rainfall. Scaife et al (2016) identify several mechanisms as to why extratropical seasonal forecast skill is most apparent in winter, and thus less apparent in summer months. Seasonal forecasts of flows in spring have only modest skill and use of historical rainfall and the HIC is recommended at both lead times.



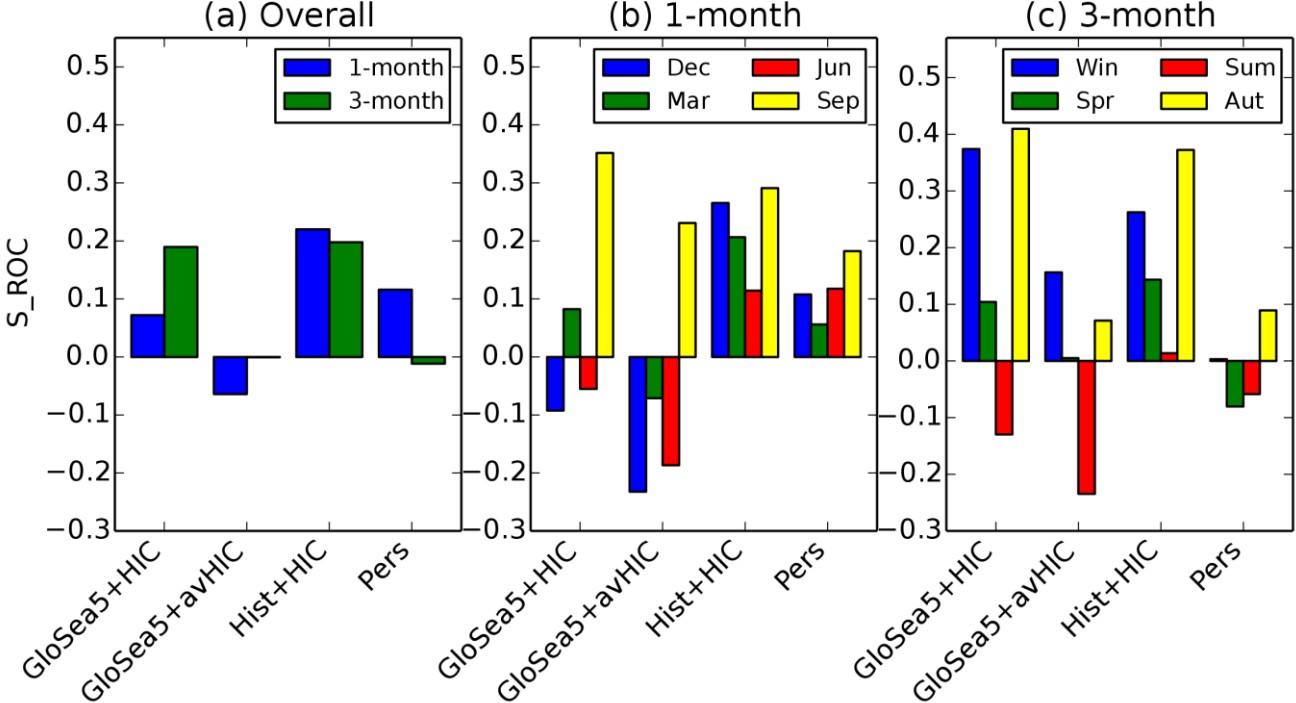

**Figure 2 Bar charts showing S$_{ROC}$ (a) averaged across three severity bands, four seasons and 17 regions, at two forecast lead times (1 and 3 months). (b,c) as (a) but presenting results for the four months/seasons separately at the two forecast lead times.**

5 Svensson et al. (2015) and Svensson (2016) highlight the value of a flow persistence forecast in Southern and Eastern regions of Britain in catchments with a high subsurface aquifer storage component and for which river flows respond slowly to rainfall. Here, analysing the ensemble results for each of the 17 regions, the skill of a flow persistence forecast in Southern and Eastern areas is apparent, but using a rainfall forecast ensemble (historical or GloSea5) and the most recent HIC is more skilful. The $S_{ROC}$ skill scores for each of the 17 regions (Figure 3), indicate that skill (averaged over all seasons) is greatly

10 dependent on the geographical region, with the historical rainfall ensemble with a HIC providing the best forecast in most regions, although at the 3-month lead time, a GloSea5 forecast ensemble with the HIC performs well.

Although an ensemble of climate forecasts can provide some indication of the range of possible rainfall totals over the next few months, ensembles of seasonal climate predictions have been shown to possess a low ratio of predictable signal to

15 unpredictable noise (Kumar 2009, Eade et al. 2014, Scaife et al. 2014). These authors indicate that a single ensemble mean forecast can have greater skill than that of the constituent ensemble members, and Murphy (1990) provides a quantification of the apparent improvement in skill through its reduction in initial state uncertainty. More recently, Eade et al. (2014) suggest that improvements in forecast skill could be achieved through using the mean of a large ensemble, followed by a post-processing step to adjust the ensemble mean so its variance agrees with the predictable component of the observed



variance. Svensson et al. (2015) restricted their UK-wide analysis of methods for winter flow forecasts to ensemble mean forecasts (3-month ahead) from GloSea5 and rainfall climatology together with a persistence forecast, and found that skilful long-range forecasts of winter flows could be achieved through a combination of the hydrogeological memory of antecedent conditions in southern and eastern parts of the UK, and from meteorological predictability in northern and western areas.

5 Here, it has not been possible to include the effect of the NAO index that was used by Svensson et al. (2015) alongside the GloSea5 winter forecasts, but the skill of ensemble mean GloSea5 forecasts has been evaluated for all seasons (not just winter). Results as Pearson correlation *(r)* are summarised for UK regions in Figure 4 for both Spring/Summer (labelled "SprSum") and Autumn/Winter ("AutWin"). The forecast methods used are labelled as in Figure 3, but for GloSea5 and Historical rainfall forecast ensembles, only the ensemble mean forecast is used.

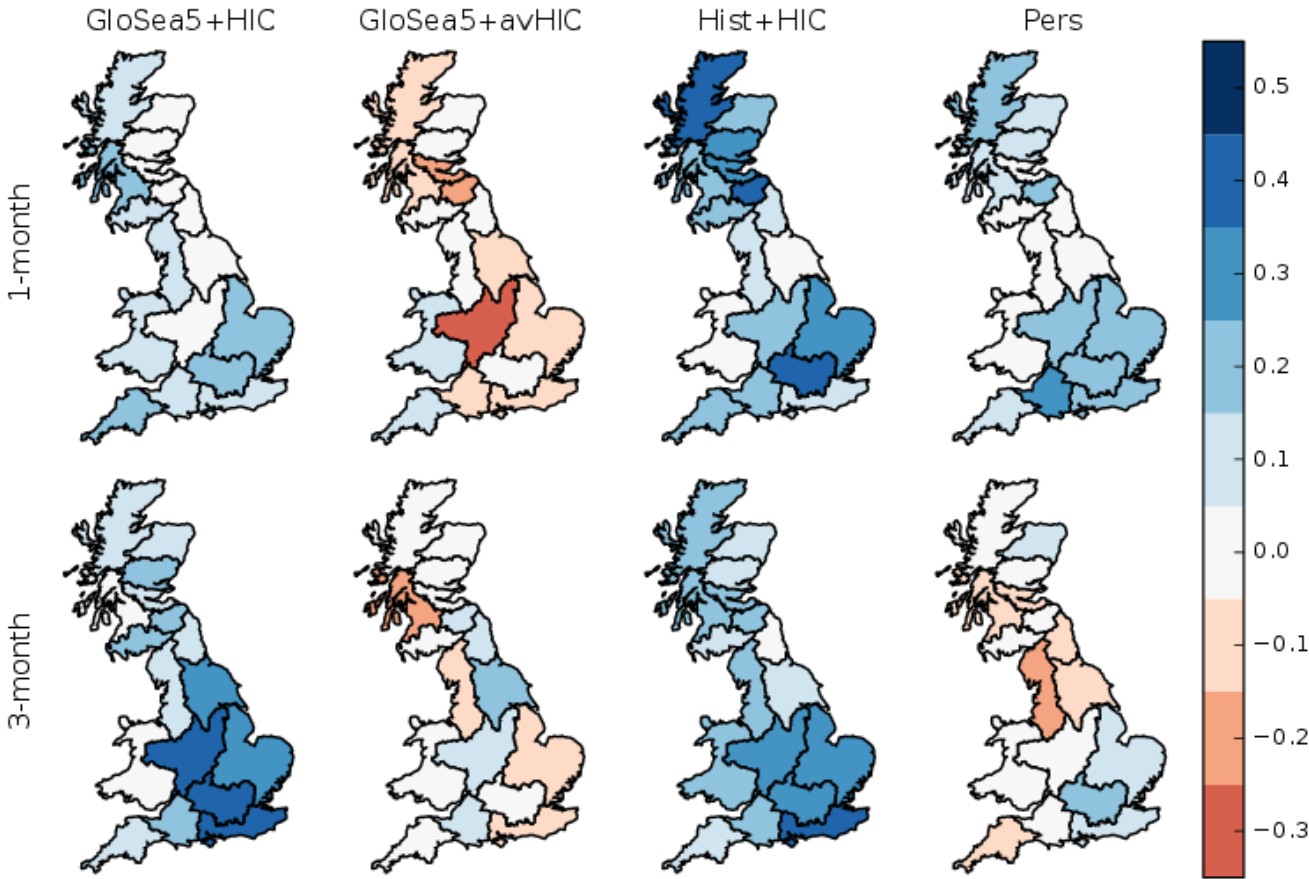

**Figure 3 UK regional maps showing mean forecast skill for four seasons in terms of the $S_{ROC}$ ensemble skill score (higher skill shown in blue)**





**Figure 4** Maps showing 1- and 3-month ahead forecast skill (correlation *r*) for Spring/Summer (top) and Autumn/Winter (bottom). Thresholds are shown for *r*<0.317 (not significant); 0.317<*r*<0.437 (significant at 5% level); 0.437<*r*<0.588 (significant at 1% level) and *r*>0.588 (significant at 0.05% level).



The difference in forecast skill between the seasons is immediately apparent, with a significant level of skill achieved for Autumn/Winter forecasts in many parts of Britain by including GloSea5 seasonal dynamical rainfall predictions:

- For Spring/Summer hydrological forecasts, some skill (at the 1% significance level in the West of Scotland) is afforded through the use of a mean historical rainfall forecast in Scotland (1-month ahead only) and Southeast regions (1- and 3-months ahead), however none of the methods tested is able to provide skilful seasonal forecasts of Spring/Summer flows in Northern and Western England or Wales. Across Britain as a whole, only the use of a mean historical rainfall forecast for a 1-month ahead flow forecasts shows any significant skill (mean correlation of 0.33, significant at the 5% level).

- By comparison, Autumn/Winter flows can be reasonably well forecast across Britain using ensemble mean rainfall forecasts based on GloSea5 or historical rainfall, with mean correlations of 0.53 and 0.50 respectively for 1-month ahead forecasts and 0.59 and 0.43 for 3-months ahead. Forecasts using historical rainfall perform better at the 1-month ahead lead-time than at 3-months ahead, and again, skill is greater in Scotland and Southeast Britain than in Wales and Northern Britain.

The use of an average HIC with Spring/Summer rainfall forecasts from GloSea5 leads to forecasts with no significant skill as it removes the main component of forecast skill which in Spring/Summer is associated with hydrological persistence. However, Autumn/Winter flow forecasts using ensemble mean GloSea5 rainfall and an average HIC perform surprisingly well across Britain, confirming that there is a significant element of skill associated with GloSea5 forecasts, particularly at the 3-month lead time.

By comparing forecast skill scores from different model configurations, it becomes possible to attribute overall forecast skill to the different model components such as HIC, GloSea5 ensemble and GloSea5 ensemble mean. Figure 5 provides an *indication* of the source of the forecast skill in Autumn/Winter for each region, alongside critical values for significance levels of Pearson's *r,* 5%, 1% and 0.05% (for a one-tailed test). For each region, the HIC skill is assumed to be the difference between the forecast skill for GloSea5 with HIC and with the long-term mean HIC ("avHIC"). Any GloSea5 skill beyond that associated with HIC can then be attributed to either the mean skill of the individual ensemble members, or to the ensemble mean forecast (if they are greater than the HIC skill).





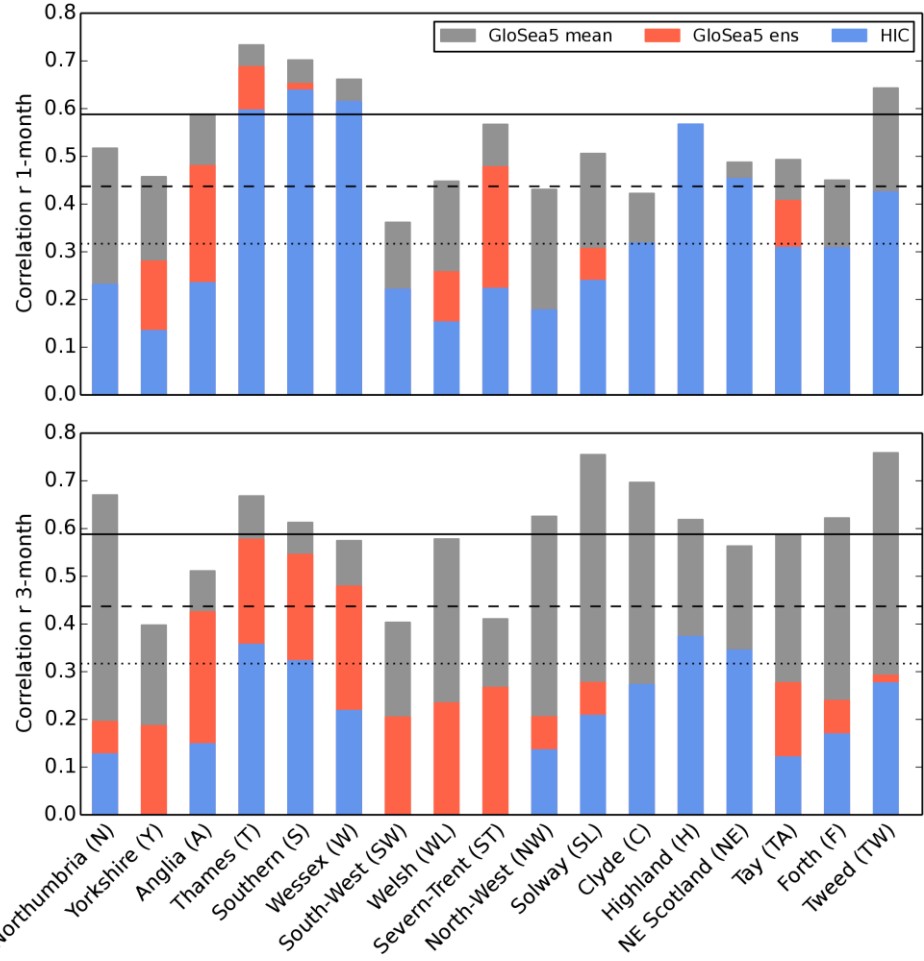

**Figure 5 Sources of flow forecast skill (Pearson's *r*) in Autumn/Winter: (a) 1-month ahead; (b) 3-months ahead. Significance levels are shown with black horizontal lines (see Figure 4).**

At the 1-month ahead lead time, the skill of Autumn/Winter GloSea5-derived forecasts in regions with long-term memory in the Southeast (Thames, Southern, Wessex) and Scotland (Highland, NE Scotland, Tweed) is primarily attributable to the dependence of the flows on the antecedent conditions provided by the HIC (blue bars in Figure 5). Averaged across all regions, the largest source (64%) of skill in the 1-month ahead seasonal flow forecasts comes from the hydrological initial condition. This component of skill is also key to the success of the historical rainfall and persistence forecasts in Autumn/Winter. In selected regions, particularly Northern and Western areas, a further 20 to 30% increase in skill arises from the GloSea5 ensemble mean (grey bars), and for a few regions (e.g. Anglia, Severn-Trent, Clyde) modest skill (10 to 20%) is derived from the mean skill of the individual ensemble members (red bars). At the 3-month ahead lead time, the influence of the HIC on forecast skill is less apparent, and only 4 regions have levels of HIC-related skill significant at the 5% level. Averaged across all regions, the HIC contributes to only a modest 30% of the forecast skill, whereas GloSea5 forecasts account for ~70% of the forecast skill, of which 46% comes from mean ensemble skill and 23% from the mean





skill of the individual ensemble members. This confirms the findings of Section 3.2 (Figure 2) that indicated that at the 3-month lead time, an ensemble of GloSea5 forecasts and a good HIC performs well. A similar analysis of the source of skill in Spring/Summer forecasts (not shown here) indicates that almost all the skill in the forecasts comes from the HIC, with little skill arising from the GloSea5 rainfall forecasts (Figure 2 indicates that an historical rainfall ensemble would be a better

choice in Spring and Summer).

**Summary and recommendations**

The Hydrological Outlook UK (HOUK, Prudhomme et al., under review) provides an insight into future hydrological conditions nationwide across Britain. It uses a range of techniques to provide likely trajectories for river flows and groundwater levels on a monthly basis, with particular focus on the next one and three months. One of the techniques uses

ensembles of UK-mean, monthly resolution seasonal rainfall forecasts provided by the Met Office GloSea5 model with hydrological modelling tools. The approach combines a high resolution, spatially distributed hydrological initial condition provided by a hydrological model (Grid-to-Grid) driven by weather observations up to the forecast time origin, with a monthly resolution water balance model (WBM) to forecast regional mean river flows for the next 1 and 3 months ahead.

The forecast skill of these regional-scale estimates of the river flows has been assessed for Britain, with results broken down between geographical regions, seasons, and at 1- and 3-month lead times. Every month, the whole ensemble of rainfall forecasts is used in the operational HOUK to provide a range (median and four quartiles) of seasonal forecast flows over the next few months. However, recent literature (discussed in Section 3.2) suggests that ensembles of seasonal climate predictions can have such a low ratio of predictable signal to unpredictable noise that the ensemble mean forecast has much

greater skill than the constituent ensemble members. Here, forecast skill has been assessed using both the whole rainfall ensemble and the ensemble mean. By comparing forecast skill scores from different model configurations, it has been possible to attribute overall forecast skill to the different model components such as hydrological initial condition (HIC), GloSea5 ensemble and GloSea5 ensemble mean.

The analysis indicates that only limited forecast skill is achievable for Spring/Summer hydrological forecasts (through the use of historical rainfall rather than rainfall forecasts), however, Autumn/Winter flows can be reasonably well forecast across Britain using ensemble mean rainfall forecasts based on either GloSea5 forecasts or historical rainfall (the preferred type of forecast depends on the region). Flow forecasts using ensemble mean GloSea5 rainfall perform the most consistently well across Britain, and provide the most skilful forecasts overall at the 3-month lead time. Most of the skill (64%) in the 1-month

ahead seasonal flow forecasts can be attributed to the hydrological initial condition, whereas for the 3-month ahead lead time, GloSea5 forecasts account for ~70% of the forecast skill.





Svensson et al (2015) highlighted that skilful seasonal predictions of UK river flows are "now a viable proposition" provided by the HOUK every calendar month at a national scale from (http://hydoutuk.net/). Currently, the whole (~42 member) ensemble of GloSea5 rainfall forecasts is used to provide a range (median and four quartiles) of seasonal forecast flows over the next few months. The enhanced level of skill that can be achieved through the use of the ensemble mean forecast alone is an important consideration, but in practice this will be very close to the ensemble median already presented alongside the minimum, maximum and mid-quartile seasonal flow forecasts. Presentation of the full range of flow scenarios for the coming 1- and 3-months is advantageous in that it alerts water managers, not only to the most likely possibility, but also to the full range of possible scenarios. It is hoped that the skill assessment presented here will lead to greater confidence in the use of HOUK seasonal flow forecasts using GloSea5, particularly in autumn and winter months,.

Despite the relatively low temporal and spatial resolution of the GloSea5 UK rainfall forecasts (currently: monthly time-step and national-scale), they can be used to provide skilful flow forecasts at a regional/national scale when combined with a hydrological model-simulated estimate of the hydrological initial condition. Given the high spatial heterogeneity in typical patterns of UK rainfall and evaporation, future development of higher resolution seasonal forecasts could lead to substantial improvements in seasonal flow forecast capability, benefitting practitioners interested in predicting flooding and water resources, not only in the UK, but potentially across Europe.

**Acknowledgements**

This work was funded by the NERC/CEH Water and Pollution Science Programme. AS was supported by the joint DECC/Defra Met Office Hadley Centre Climate Programme (GA01101), the UK Public Weather Service research programme.

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
