# Peer review of "A national-scale seasonal hydrological forecast system: development and evaluation over Britain"

_Hydrology and Earth System Sciences, 2017_

## Referee Comment (RC1) · Anonymous Referee #1 · 22 May 2017

The manuscript illustrates the development and the evaluation of a national-scale seasonal hydrological forecast system over Britain. A high resolution hydrological model is used to estimate the initial conditions of a simple water balance model used to forecast regional seasonal flows over Britain. As meteorological forcings seasonal re-forecasts from GloSea5 and historical rainfall are used. As additional reference for verification flow persistence is used.

General: The paper is well written, the methodology and results are nicely presented and compared. The real value of this study is the combination of a high resolution hydrological model to estimate the initial conditions and a simple water balance model which fits better to the meteorological seasonal forecast input data than a high resolu-

tion model. The regional WBM performance using observed data and the forecasting skill show promising results. The paper should be foreseen for publication in HESS after minor revisions.

Comments: The objective of the seasonal forecasting system is not really clear. From the introduction I assume the focus on flood forecasting, from the forecast variables, regional mean flow 1-month / 3-month ahead, I assume the focus on drought forecasting. Please specify! If the focus is flood forecasting, the relationship between regional mean flow and relevant flood properties (peak, volume,...) should be shown to demonstrate the potential flood predictability of the system. p 7 l 15: What is UK mean SAAR? Please add explanation / reference. p 12 l 6: do you use ensemble mean of GloSea5 / historical rainfall 1-month / 3-month ahead as input for WBM or do you use the ensemble mean of the WBM seasonal flow forecasts? What is the difference in skill between the two ensemble mean forecasts (rainfall ensemble mean as input vs. flow ensemble mean)?

---

## Referee Comment (RC2) · Anonymous Referee #2 · 2 Jun 2017

Review

This manuscript explores the potential for 1 month and 3 month hydrologic forecasts on a national scale, for different seasons. As a precursor, the manuscript describes how high resolution spatial information from a hydrologic model can be used to estimate initial conditions in a simplified manner.

General: Overall, the paper is well written and the structure provides for a coherent progression through the various sections. The main takeaway points, relative to the stated intention of development an evaluation of the system, are clear in that the seasonal differences in forecast skill are prominently noted and discussed. However, as the

manuscript as identifies the desire to link to decision making, I would suggest clarifying that link, maybe providing a section, as slight mentions of this link are unconvincing. I recommend the manuscript for publication in HESS after addressing minor revisions.

Page 2 Line 5-7 What is meant by perceived lack of skill? Is there skill or not? I would clarify as the lack of skill could discourage development of forecasts or it can also be the case that there is skill, yet amongst key people making decisions of research priorities, it could be perceived to not have skill. The second condition is more complicated and likely will need to be resolved with social science.

Line 26 A general note overall, and referencing the potential value for practitioners in line 29 of the abstract, it is important not to state skill acceptability in an overarching sense. In the literature differences in both perception of skill and more importantly acceptable levels of skill in order to justify using that forecast can vary across sectors and will likely be different for users compared to forecast developers. For more on this see: Hartmann, H. C., Pagano, T. C., Sorooshian, S., & Bales, R. (2002). Confidence builders: Evaluating seasonal climate forecasts from user perspectives. Bulletin of the American Meteorological Society, 83(5), 683-698.

Page 3 Line 16 – Correct in stating that downscaling would be necessary, but is there any evidence that downscaling would be a 'worthwhile' activity for improving national estimates of water flows?

Line 17 – What is meant by realistic water flows? I suggest adding context or changing the word as, in the current form, this is at unnecessarily high risk for misinterpretation.

Page 4 Line 19 – I suggest clarifying the period in which the long term average is calculated over. I think it can be interpreted as monthly or seasonally, which could impact the result.

Page 7 Line 14 Does 'relative to the rainfall forecast climatological mean' imply the 3 month anomaly will be distributed over each month based on a month's relative mean

contribution to the total seasonal precipitation?

Page 10 Line 1 The Forth region has median model performance for 1 month or 3 month leads? Or both combined?

Line 14 Seems that a comma is in error or a capital T is in error

Line 16 2a shows persistence on a 1 month lead more skillful than GloSea5+HIC. Can you explain why? Or I suggest noting that for overall assessments on a 1 month lead, persistence forecast should be explored in more depth.

Page 12 Line 15 Agreed, yet this point (how inclusion of avHIC with GloSea5 lead to 0 skill during some seasons) presents another that may be worth addressing – In figure 3, there exists an interesting pattern of skill at the sub-national level. Any thoughts on why?

Page 15 Figure 5 Is it possible to include a map that labels the regions (abbreviations should suffice)? I think the paper as a whole would benefit from this, even if one of the maps used in a previous figure could do this.

Line 9 It would be useful to note which regions are included in the 'Northern' and 'Western' cluster.

Page 17 Line 7 The presentation of this information to any user does not inherently alert them to anything. If they perceive the information (both the most likely possibility and the full range) to be trustworthy and if they are able to justify making (or changing a from normal protocol) a decision based on that information, then I can justify the use of alert in this context. I would suggest, in an over simplistic manner, what the presentation of the full range of scenarios does – presents the full range and the mean. This could be useful and interesting (and then, maybe advantageous) for a user but only if it is perceived to be relevant by them. Also this sentence uses the phrase 'presentation of the full range of scenarios' twice, so I would suggest re wording even if the above suggestion on content is not followed

Page 17 Line 14-16

Although the section notes recommendations will be included, I do not find any except for a weak statement regarding how an increase in spatial resolution could lead to improvement. This is not a new finding and I would consider reflecting on the content of the paper to develop recommendations that are more relevant.

It may be interesting to explore the role of ENSO. Referencing 4.3.5 of van Oldenborgh et al 2005, there is potential for some skill from ENSO for parts of the UK, including Scotland. Noting the skill in Scotland (figure 3), it could be a useful exercise to disaggregate by ENSO phase both in the target month and in the month of forecast issuance.

Jan van Oldenborgh, G., Balmaseda, M. A., Ferranti, L., Stockdale, T. N., & Anderson, D. L. (2005). Did the ECMWF seasonal forecast model outperform statistical ENSO forecast models over the last 15 years?. Journal of climate, 18(16), 3240-3249.
* * *

---

## Author Response (AR1)

**Response to comments from the Editor and reviewers #1 and #2**

Authors' responses in red.

**Response to Anonymous Referee #1**

The manuscript illustrates the development and the evaluation of a national-scale seasonal hydrological forecast system over Britain. A high resolution hydrological model is used to estimate the initial conditions of a simple water balance model used to forecast regional seasonal flows over Britain. As meteorological forcings seasonal re-forecasts from GloSea5 and historical rainfall are used. As additional reference for verification flow persistence is used.

General:
The paper is well written, the methodology and results are nicely presented and compared. The real value of this study is the combination of a high resolution hydrological model to estimate the initial conditions and a simple water balance model which fits better to the meteorological seasonal forecast input data than a high resolution model. The regional WBM performance using observed data and the forecasting skill show promising results. The paper should be foreseen for publication in HESS after minor revisions.

Thank you for the positive comments.

Comments:
The objective of the seasonal forecasting system is not really clear. From the introduction I assume the focus on flood forecasting, from the forecast variables, regional mean flow 1-month / 3-month ahead, I assume the focus on drought forecasting. Please specify! If the focus is flood forecasting, the relationship between regional mean flow and relevant flood properties (peak, volume,...) should be shown to demonstrate the potential flood predictability of the system.

The Abstract, Introduction and Summary will be amended to make this clearer. The forecasts issued provide seasonal mean river flows over the next 1 and 3 months (not flood peaks). The objective is to provide a wide audience, ranging from water managers to members of the public, with insight into whether river flows might be expected to be above, below, or in the normal range over the next few months based on the best information available.

p 7 l 15: What is UK mean SAAR? Please add explanation / reference.

SAAR is Standard-period Average Annual Rainfall (mm) (e.g. http://nrfa.ceh.ac.uk/rainfall-statistics). However, we will amend the manuscript to say "average annual rainfall", instead of "SAAR". We created annual mean rainfall for a 49-year period that is longer than a standard period of 30 years, so we should not have used the acronym "SAAR" in the first place.

p 12 l 6: do you use ensemble mean of GloSea5 / historical rainfall 1-month / 3-month ahead as input for WBM or do you use the ensemble mean of the WBM seasonal flow forecasts?

We intended to say that the ensemble mean GloSea5 rainfall forecast was used as input to the WBM. The manuscript will be amended to say "the skill of ensemble mean GloSea5 rainfall forecasts has been evaluated for all seasons…".

What is the difference in skill between the two ensemble mean forecasts (rainfall ensemble mean as input vs. flow ensemble mean)?

We have not evaluated the skill in the ensemble mean flow forecast, just the skill in the flow forecast using an ensemble mean rainfall forecast. I would expect these two variants to be very similar, if not identical, as the equations are all linear (i.e. construction of the mean and the WBM are linear). It is also quicker to make one flow forecast from an ensemble mean rainfall forecast, rather than average an ensemble of flow forecasts.

**Response to Anonymous Referee #2**

This manuscript explores the potential for 1 month and 3 month hydrologic forecasts on a national scale, for different seasons. As a precursor, the manuscript describes how high resolution spatial information from a hydrologic model can be used to estimate initial conditions in a simplified manner.

General: Overall, the paper is well written and the structure provides for a coherent progression through the various sections. The main takeaway points, relative to the stated intention of development an evaluation of the system, are clear in that the seasonal differences in forecast skill are prominently noted and discussed. However, as the manuscript as identifies the desire to link to decision making, I would suggest clarifying that link, maybe providing a section, as slight mentions of this link are unconvincing. I recommend the manuscript for publication in HESS after addressing minor revisions.

Thank you.
With the aim of clarifying links to decision-making, extra sentences will be added to the manuscript (in the conclusions) to say:
"*The HOUK has been in operation for 4 years (publically available from autumn 2013) and thus is a relatively new product. At present, automated web statistics indicate approximately 300 readers or users of the HOUK website per month (Prudhomme et al., under review). Exactly how water managers use the HOUK in practice has not yet been assessed, but ongoing evaluations of the skill in the different methods used in the construction of the Outlook will undoubtedly help provide the evidence required to support use of the product in decision-making.*"

Specific comments:
Page 2 Line 5-7 What is meant by perceived lack of skill? Is there skill or not? I would clarify as the lack of skill could discourage development of forecasts or it can also be the case that there is skill, yet amongst key people making decisions of research priorities, it could be perceived to not have skill. The second condition is more complicated and likely will need to be resolved with social science.

This sentence will be re-worded to emphasise that published literature as recently as 2011 indicated that seasonal forecasts of rainfall in extratropical areas (such as Britain) had little skill at lead times of more than 1-month. Thus there was little incentive to use the forecasts to support decision-making.

Line 26 A general note overall, and referencing the potential value for practitioners in line 29 of the abstract, it is important not to state skill acceptability in an overarching sense. In the literature differences in both perception of skill and more importantly acceptable levels of skill in order to justify using that forecast can vary across sectors and will likely be different for users compared to forecast developers. For more on this see: Hartmann, H. C., Pagano, T. C., Sorooshian, S., & Bales, R. (2002). Confidence builders: Evaluating seasonal climate forecasts from user perspectives. Bulletin of the American Meteorological Society, 83(5), 683-698.

*This is a good point. The sentence in the abstract will be clarified to say "Given the high spatial heterogeneity in typical patterns of UK rainfall and evaporation, future development of skilful spatially distributed seasonal forecasts could lead to substantial improvements in seasonal flow forecast capability, potentially benefitting practitioners interested in predicting hydrological extremes, not only in the UK, but also across Europe."*

Page 3 Line 16 – Correct in stating that downscaling would be necessary, but is there any evidence that downscaling would be a 'worthwhile' activity for improving national estimates of water flows?
The sentence will be re-worded to make it clear that we aim to provide flow estimates nation-wide, rather than at a national scale:
*"… seasonal rainfall forecasts do not provide detailed weather information at this resolution and would typically require spatio-temporal downscaling to achieve good estimates of river flow for catchments or regions nation-wide".*

Line 17 – What is meant by realistic water flows? I suggest adding context or changing the word as, in the current form, this is at unnecessarily high risk for misinterpretation.

We have changed the word "realistic" to "good". Please see re-worded sentence above.

Page 4 Line 19 – I suggest clarifying the period in which the long term average is calculated over. I think it can be interpreted as monthly or seasonally, which could impact the result.

The sentence will be clarified to say "long term monthly average".

Page 7 Line 14 Does 'relative to the rainfall forecast climatological mean' imply the 3 month anomaly will be distributed over each month based on a month's relative mean contribution to the total seasonal precipitation?

Yes. The sentence can indeed be simplified to say *"Disaggregation of the 3-month ahead forecast into monthly rainfall amounts is achieved through distributing the 3-month rainfall forecast anomaly between the 3 individual months according to their relative contribution to the UK mean seasonal rainfall (1962 to 2010)."*

Page 10 Line 1 The Forth region has median model performance for 1 month or 3 month leads? Or both combined?

This is for the 1-month lead time and the text has been clarified.

Line 14 Seems that a comma is in error or a capital T is in error

Ah, thank you. The offending comma will be changed to a full stop.

Line 16 2a shows persistence on a 1 month lead more skillful than GloSea5+HIC. Can you explain why? Or I suggest noting that for overall assessments on a 1 month lead, persistence forecast should be explored in more depth.

The sentences will be clarified as follows: "*At the 1-month lead time the WBM with G2G HIC driven by an historical rainfall (climatology) ensemble performs best, and the forecasts based on persistence or Glosea5+HIC perform less well, but show some skill. For the longer 3-month lead time, the WBM with G2G HIC driven by either historical or GloSea5 rainfall perform similarly; persistence forecasts (Pers) or use of an average HIC are not recommended at this lead time.*"

The explanation for the skill in persistence forecasts is addressed in the regional skill breakdown in the next but one paragraph in the manuscript.

Page 12 Line 15 Agreed, yet this point (how inclusion of avHIC with GloSea5 lead to 0 skill during some seasons) presents another that may be worth addressing – In figure 3, there exists an interesting pattern of skill at the sub-national level. Any thoughts on why?

This probably refers to p14 Line 15. Yes, the apparent skill in use of an average HIC with GloSea5 in Figures 3 and 4 is interesting. Figure 4 indicates that these forecasts are only skilful in Autumn and Winter, and particularly at the 3-month ahead lead time. The regions where the skill is greatest (Northern and Western regions, excluding Northern Scotland) are areas with less subsurface storage than the SouthEast regions and Northern Scotland, and thus where persistence forecasts are less skilful. We would speculate that in these areas the skill arises from the improved skill in GloSea5 at the 3-month lead time, coupled with less dependence on a good HIC in these areas.

The sentence in the manuscript will be changed to say:
"*…Autumn/Winter flow forecasts using ensemble mean GloSea5 rainfall and an average HIC perform surprisingly well across Britain, confirming that there is a significant element of skill associated with GloSea5 forecasts in Autumn/Winter at the 3-month lead time, often resulting in skilful flow forecasts in regions where this skill is less dependent on a good HIC.*"

Page 15 Figure 5 Is it possible to include a map that labels the regions (abbreviations should suffice)? I think the paper as a whole would benefit from this, even if one of the maps used in a previous figure could do this.

A map labelling the regions is already included in Figure 1b. It is rather small, and we will make it larger in the revised manuscript. The Figure 5 caption will be changed to reference this map.

Line 9 It would be useful to note which regions are included in the 'Northern' and 'Western' cluster.

The text will be amended to list the regions (Northumbria, Yorkshire, South-West, Welsh, North-West, Solway, Clyde, Forth, Tweed)

Page 17 Line 7 The presentation of this information to any user does not inherently alert them to anything. If they perceive the information (both the most likely possibility and the full range) to be trustworthy and if they are able to justify making (or changing a from normal protocol) a decision based on that information, then I can justify the use of alert in this context. I would suggest, in an over simplistic manner, what the presentation of the full range of scenarios does – presents the full range and the mean. This could be useful and interesting (and then, maybe advantageous) for a user but only if it is perceived to be relevant by them. Also this sentence uses the phrase 'presentation of the full range of scenarios' twice, so I would suggest re wording even if the above suggestion on content is not followed

This is useful feedback. The text will be modified to say *"Continued presentation of this full range of flow scenarios for the coming 1- and 3-months may be advantageous in that it informs water managers, not only of the most likely possibility, but also to the range of possibilities."*

Page 17 Line 14-16
Although the section notes recommendations will be included, I do not find any except for a weak statement regarding how an increase in spatial resolution could lead to improvement. This is not a new finding and I would consider reflecting on the content of the paper to develop recommendations that are more relevant.

This is a good point. An additional short paragraph will be added to provide recommendations:
"*Based on the skill analysis presented here, users of the Hydrological Outlook UK would be advised to have greatest confidence in Autumn and Winter flow forecasts that use GloSea5 rainfall, particularly at the 3-month lead time. For Spring/Summer flow forecasts, use of an ensemble forecast based on historical rainfall is surprisingly good and would be recommended for use across Scotland, and flow forecasts based on persistence were found to be the most skilful in South-East regions (Thames, Anglia, Wessex and Southern)."*

It may be interesting to explore the role of ENSO. Referencing 4.3.5 of van Oldenborgh et al 2005, there is potential for some skill from ENSO for parts of the UK, including Scotland. Noting the skill in Scotland (figure 3), it could be a useful exercise to disaggregate by ENSO phase both in the target month and in the month of forecast issuance.
Jan van Oldenborgh, G., Balmaseda, M. A., Ferranti, L., Stockdale, T. N., & Anderson, D. L. (2005). Did the ECMWF seasonal forecast model outperform statistical ENSO forecast models over the last 15 years? Journal of climate, 18(16), 3240-3249. Interactive comment on Hydrol. Earth Syst. Sci. Discuss., https://doi.org/10.5194/hess-2017-154, 2017.

This is an interesting link to make, thank you for the reference. This approach may well be worth exploring in another piece of work, but is beyond the scope of the current manuscript.

**Editor Decision: Publish subject to minor revisions (further review by Editor) (07** Jul 2017) by Andy Wood
Comments to the Author:
In general, the authors' proposed revisions appear adequate to address the concerns and comments raised by the reviewers, and I encourage the authors to adjust the manuscript along the lines proposed and resubmit for my further review.

Authors response: Thank you, these revisions are now in hand.

In addition, I wish to add the following comments for the authors to consider in their revision.

Major comment:
I'm struck by the relative arbitrariness of the approach in which HICs are drawn from a high-res land surface model to initialize seasonal ensemble forecasts that are propagated forward with a conceptual scheme at the opposite end of the spectrum in terms of complexity (ie the WBM). Are there examples of this type of approach being used elsewhere that could be referenced?

Authors response: Other studies referenced in the manuscript (p2, Introduction) highlight the importance of the hydrological initial condition (HIC) to seasonal hydrological forecast skill, and our aim with the WBM formulation was to combine the high resolution, most recent, HIC available from the G2G with a simple monthly rainfall forecast from GloSea5. It is not only HICs that are extracted from the high-resolution G2G hydrological model, other spatial grids derived from G2G output (such as Smax, Smin, S_bar, q_mean) are used by the WBM. We are unaware of other studies that have used a scheme of the type presented in this manuscript, and would welcome hearing of any that we have missed.

The given rationale for the strategy is that there's no appreciable value to downscaling climate forecasts to the scale of the G2G model, yet there have been numerous examples of seasonal forecasting in which coarse climate forecasts are in fact downscaled to resolutions on the order of 10K or finer, and used to drive flow forecasts.

Authors response: To develop a seasonal hydrological forecasting approach for semi-operational use, a method was required which would make the best use of the available coarse resolution GloSea5 rainfall forecasts alongside routinely available hydrological information. Early modelling trials (unpublished) explored using a 30-year ensemble of historical rainfall observations to produce a spatially-variable, daily ensemble of 1140-member G2G-derived rainfall forecasts downscaled from national-scale to 1km resolution. Although the approach showed some promise, detecting and interpreting any forecast signal in the resulting forecast "noise" was not straightforward, and the possibility of developing a simpler, more elegant approach that could be explained clearly to a wide audience was explored in preference (this manuscript). In place of dynamic downscaling to 1km we have downscaled in space only, using climatological mean UK monthly rainfall patterns on a 1km grid to downscale GloSea5 (a similar approach was used by Bell et al., (2009) to downscale 25km RCM rainfall data to a 1km resolution grid). We have clarified the description of the spatial downscaling method applied in Section 2.3: "*This approach to spatial-downscaling using historical mean rainfall observations is similar to one used by Bell et al. (2009) to downscale 25km-resolution regional climate model data to a 1km resolution.*"

Is another practical rationale that the computational cost of running the G2G model ensembles with downscaled forecasts is not justified by any added skill from to computing a full-resolution seasonal flow forecast after downscaling?

Authors response: The added computational cost of combining multiple downscaling scenarios (in time and space) with a 38-member Glosea5 forecast ensemble is an important consideration, but was not the main one.

In any case, I think this somewhat bespoke/uncommon approach to seasonal regional forecast construction is an interesting choice that warrants a bit more motivation.

Authors response: We will amend the manuscript, particularly the motivation for the development of the WBM method (p3, lines 15-27) as follows: "*While rainfall downscaling is relatively straightforward for a particular location or catchment, using national-scale monthly rainfall forecasts to produce pixel-scale daily rainfall would require an ensemble downscaling approach based on either a weather generator or historical analogues, generating large multiples of ensemble flow forecasts. This approach has been explored in other studies (e.g. Manzanas et al. 2017, Charles et al. 2012), which showed that dynamic or statistical downscaling of seasonal forecasts can reduce local biases in variables such as temperature or rainfall, but do not necessarily improve the overall forecast skill.*".

And is there is any way to benchmark it against any alternatives that would provide practical insights? For instance, for regional prediction of flow anomalies, is there added value from taking HICs from G2G versus from a long-term run of a simple conceptual monthly model -- could that be tested? And is there any way to assess the marginal value of this approach, versus other alternatives (such as the far more common use of one model for both HICs *and* forecast, such as at SMHI or in the US NWS), against actual obs flows -- perhaps for a handful of case study basins where such flow estimates are available? I realize that this may all lie beyond the scope of this paper. In any case a broader discussion of the rationale for this particular prediction scheme, and acknowledgement of other alternatives, could add value to the paper.

Authors response: The question about benchmarking the G2G/WBM method against alternative sources of HIC is a relevant one. We have started looking into this already, but this is ongoing research and out of scope for the current manuscript.

I find the paper's attribution of forecast skill using the component-on/component-off approach very useful and effective, as a technique that is quite common in seasonal climate prediction (ie, use a climatological Indian Ocean or continental SM, versus a dynamic one), but less common in hydrologic prediction studies. It provides a slightly different angle amidst the recent interest in the attribution of seasonal hydrologic prediction skill.

Minor comments:
p8, l8 -- 'Clim' doesn't match the 'Hist' term later used in Fig 2 and elsewhere.

Authors response: Now corrected, we meant to say "Hist". Thank you for spotting that.

p6, l10 – the text states: "The coarse spatial resolution of the input rainfall forecasts has discouraged the development of river flow forecasts at a 1km resolution, and production of regional scale forecasts (in preference to national scale) is viewed as a pragmatic compromise. " Is that the sole reason or is the computational cost of running a 1 km resolution model for seasonal ensembles also a limiting factor?

Authors response: Computational cost is a factor discouraging but not preventing use of the 1km resolution G2G with hundreds of ensemble members (though the linux-cluster would allow for multi-ensemble model runs if necessary). The seasonal flow forecasts from the monthly WBM are available at a 1km resolution (Section 2,2 final paragraph) before regional averages are computed, but the 1km WBM forecasts have not been assessed and are not used in practice. The methods developed here are part of a semi-operational "Hydrological Outlook" providing UK-wide seasonal forecasts of river flows to operational agencies and the general public. Using a national-scale mean rainfall product to provide a semi-operational flow forecast at a 1km resolution, when the rainfall information is simply not available at such a high resolution, seemed misleading – the provision of a regional forecast seemed more appropriate. The manuscript text will be clarified as follows: "*The coarse spatial resolution of the input rainfall forecasts has discouraged the development of river flow forecasts at a 1km resolution to ensure that users of the Hydrological Outlook do not infer that rainfall forecasts are available or skilful at this resolution. Production of regional scale forecasts (in preference to national scale) is viewed as a pragmatic compromise*".

p6, l5 – when/why does it arise that the Qm_bar/Sm_bar >= 1? More explanation or insight would be helpful – especially as these are climatological averages. The monthly runoff ratio

should almost always be between 0 and 1 unless a snowpack-related delay in runoff timing is involved.

Authors response: This is a good question, and as a result of this we have amended the preceding explanation and notation to explain the model more clearly, and to make it clear that qm_bar is G2G-derived river flow per unit catchment area, and not runoff. Across Britain qm_bar <= Sm_bar in most places, but for a few highly spatially variable catchments, and in wetter months, there will be some situations where qm_bar > Sm_bar. A note has been made in the text to say this could occur in areas of high spatial variability.

[revised manuscript text omitted]

---

## Author Response (AR2)

Editor Decision: Publish subject to technical corrections (14 Aug 2017) by Andy Wood

Comments to the Author:

Although the review process raised several suggestions for additional scope to strengthen the paper (e.g., benchmarking the hi-res/low-res hybrid approach versus more traditional variations), and these were not taken, the paper in its current form nonetheless presents useful analyses and appears to be suitable for publication.

I suggest a careful proofreading before submitting a final version to ensure clarity in all of the passages highlighted by the reviewers, after the changes that have been added in this latest revision. It is also customary to label colorbars and provide their units in a plot (eg Figs 3 & 4), rather than in the caption.

Authors Response:

We'd like to thank the editor for advice leading to an improved manuscript.

We have proof-read carefully, making minor changes to punctuation, and have added legend labels to Figures 3 and 4 (the performance measures have no units).